# Stability Study of Isoniazid and Rifampicin Oral Solutions Using Hydroxypropyl-Β-Cyclodextrin to Treat Tuberculosis in Paediatrics

**DOI:** 10.3390/pharmaceutics12020195

**Published:** 2020-02-24

**Authors:** Ana Santoveña-Estévez, Javier Suárez-González, Amor R. Cáceres-Pérez, Zuleima Ruiz-Noda, Sara Machado-Rodríguez, Magdalena Echezarreta, Mabel Soriano, José B. Fariña

**Affiliations:** 1Departamento de Ingeniería Química y Tecnología Farmacéutica, Facultad de Farmacia, Universidad de La Laguna, 38200 La Laguna (Tenerife), Spain; jsuarezg@ull.edu.es (J.S.-G.); amorraycocaceresperez1996@gmail.com (A.R.C.-P.); zule.rn@gmail.com (Z.R.-N.); alu0100760705@ull.edu.es (S.M.-R.); mechezar@ull.edu.es (M.E.); msoriano@ull.edu.es (M.S.); jbfarina@ull.edu.es (J.B.F.); 2Instituto Universitario de Enfermedades Tropicales y Salud Pública de Canarias (IUETSPC), Universidad de La Laguna, 38200 La Laguna (Tenerife), Spain; 3Programa Doctorado en Ciencias de la Salud. Universidad de La Laguna, 38200 La Laguna (Tenerife), Spain; 4Programa de Doctorado en Ciencias Médicas y Farmacéuticas, Desarrollo y Calidad de Vida. Universidad de La Laguna, 38200 La Laguna (Tenerife), Spain

**Keywords:** first-line antituberculosis paediatric treatment, dose combination, isoniazid, pyrazinamide, rifampicin

## Abstract

(1) Background: First-line antituberculosis treatment in paediatrics entails the administration of Isoniazid, Pyrazinamide, and Rifampicin. This study examines the possibility of developing a combined dose liquid formulation for oral use that would facilitate dose adjustment and adherence to treatment for younger children. (2) Methods: The active pharmaceutical ingredients stability under in vitro paediatric digestive pH conditions have been checked. The samples were studied as individual or fixed combined paediatric dosages to determine the pH of maximum stability. The use of hydroxypropyl-β-cyclodextrin to improve Rifampicin solubility and the use of ascorbic acid to increase the stability of the formulation have been studied. (3) Results: Maximum stability of combined doses was determined at pH 7.4, and maximum complexation at pH 8.0. Taking this into account, formulations presented the minimum dose of two active pharmaceutical ingredients dissolved. The addition of ascorbic acid at 0.1% *w*/*v* enables the detection of a higher remaining quantity of both drugs after three days of storage at 5 °C. (4) Conclusions: a formulation which combines the minimum paediatric dosages dissolved recommended by WHO for Isoniazid and Rifampicin has been developed. Future assays are needed to prolong the stability of the formulation with the aim of incorporating Pyrazinamide to the solution.

## 1. Introduction

First-line treatment of drug-sensitive tuberculosis (TB) consists of a two-month intensive phase with Isoniazid (INH), Pyrazinamide (PZA), and Rifampicin (RFP), followed by a continuation phase with INH and RFP for at least four months [1]. In the case of adults, a fixed-dose combination (FDC) of these three active pharmaceutical ingredients (APIs) is marketed as a sugar-coated tablet [2]. The combination of these three antiTBs in a solid oral tablet as FDC has many advantages over their individual administration, e.g., it improves adherence to treatment and reduces dosage errors [3]. A FDC for paediatrics has recently been approved by WHO/SRA/ERP as a dispersible (non-coated) tablet with a strawberry flavour at 50, 150, and 75 mg for INH, PZA, and RFP, respectively [4]. These tablets were developed in cooperation with TB Alliance and MacLeods [5] who have incorporated these APIs with the correct paediatric doses revised by WHO in 2010 [6]. However, this FDC has certain disadvantages: the doses are established according to a very wide weight range [3], tablets incorporate excipients that are not suitable for paediatrics (aspartame, microcrystalline cellulose), and quality problems as the poor bioavailability of RFP. The main reason hypothesized in the literature regarding this last question is the interaction between INH and RFP to yield isonicotinyl hydrazine (HYD) [7,8]. RFP is hydrolysed under acidic conditions to three-formylrifamycin, which reacts with INH to form HYD. HYD converts back to INH and three-formylrifamycin. As a result, INH is recovered, but RFP is lost [9]. Other reasons that result in FDC quality problems studied in the literature are the characteristics of the raw material, changes in crystalline form, excipients, manufacturing and/or process variables, variability in absorption and metabolism, etc. [10].

Piñeiro et al. in the 2016 study titled “Recommendations for the preparation and administration of antituberculosis drugs in children” referred to the need of a compounding oral liquid formulation which combines the APIs of first-line treatment. This oral liquid formulation would be better accepted in paediatrics and would allow greater flexibility in determining the volume of doses to be administered per kilogram of body weight.

In order to elaborate a liquid formulation, the solubility and every API dose to be added will determine the type of dispersed system, homogeneous (solution), or heterogeneous (suspension). In order to ensure the administration of the right API content in each dose, a solution is preferred, more so in the case of combining several drugs in the same liquid formulation. However, this is difficult to obtain because INH, PZA, and RFP have different solubilities and permeabilities and are classified in different classes of the Biopharmaceutical Classification System (BCS). The first two belong to Class I, high solubility and high permeability, while the last one is the most problematic and is classified as class IV in the BCS due to its low solubility (pH-dependence) and low permeability [11].

The problem regarding the solubility of RFP could be overcome by the formation of inclusion compounds with cyclodextrins, as it has been already studied by hydroxypropyl-β-cyclodextrin (HPBCD) [12,13,14]. In 2004, Alves et al. analyzed the molecular association of RFP with HPBCD at pH 6.9 and different ionic strengths. They checked that the phase solubility behaviors showed an AL type diagram with a stoichiometry of molecular complex probably 1:1, and an increase in the aqueous solubility depending on the ionic strength [12]. The formulation of RFP:HPBCD complexes for lung nebulization has been studied by Tewes et al. [13]. The best complexation efficiency was obtained at pH 9 and then extemporaneously buffered at pH 7.4 for their nebulization in the lung. This formulation allowed higher RFP dosing into the lung by increasing its solubility. More recently, other authors checked solubility, stability, and antibacterial activity of the inclusion complex formed and concluded that RFP:HPBCD complex might be a promising system for oral or parenteral drug delivery to treat bacterial infections [14]. However, the use of cyclodextrins has not been sufficiently focused on paediatric treatment. Despite the fact that there are cyclodextrins like HPBCD, which are safe for children under two years of age according to the EMA, it is authorized for all routes of administration except for nasal [15,16].

In this study, the stability of the three APIs in liquid form was studied both individually or in a fixed combination of paediatric doses, simulating the in vitro conditions of the gastrointestinal tract of children. Then, a solubility diagram was used to study the influence of the HPBCD over the solubility of RFP at the pH of in vitro stability. The final goal was to discover if this stability allows us to develop the most adequate liquid formulation, combining at least two APIs (INH and RFP) with adequate excipients for children for oral administration at continuation or intensive phase of paediatric TB treatment as an alternative when no marketed drugs are available.

## 2. Materials and Methods

The following antiTB used were: Isoniazid (INH, Acofarma, Madrid, Spain), Pyrazinamide (PZA, Sigma Aldrich, Darmstadt, Germany), and Rifampicin (RFP, Sigma Aldrich, Darmstadt, Germany). Cyclodextrin was HP-β-cyclodextrin (HPBCD, Kleptose^®^, Rockette Frères, Beinheim, France). Others materials included were: Sodium chloride (Sigma Aldrich, Darmstadt, Germany), pepsin (Sigma Aldrich, Darmstadt, Germany), hydrochloric acid (Merck, Darmstadt, Germany), sodium hydroxide (Panreac, Barcelona, Spain) pancreas powder (Escuder, Barcelona, Spain), and dihydrogen potassium phosphate (Merck, Darmstadt, Germany).

### 2.1. Ultra Performance Liquid Chromatography (UPLC) Method

The antiTBs were analyzed by reversed phase UPLC in an Acquity UPLC^®^ H-Class System (Waters, Milford, MA, USA) using an adapted High Performance Liquid Chromatography (HPLC) method [17,18]. The data acquisition software was Astra 6.0.1. (Chromatographic Manager, Waters Corporation). The column used was X-Select^®^ CSMTM C18 (2.1 × 75 mm; 2.5 µm). Two isocratic methods based on an acetonitrile/phosphate buffer 0.05 mM pH 3.7 mixture were used to analyze the three APIs, changing the solvent proportions: 2/98 *v*/*v* (method 1, for INH and PZA), and 38/62 *v*/*v* (method 2, for RFP). The flow rate was 0.5 mL/min, the injection volume was 10 µL, and the UV detection was carried out at 254 nm. Chemicals and reagents were UPLC grade. All samples and solvents were filtered with 0.2 µm pore-size filters (Millipore, Billerica, MA, USA) and degassed. The method has been validated in a previous work [19] and is a simple stability-indicating method, which allowed us to detect and quantify the first-line drugs used in TB treatment as pure patterns and after its extraction from formulations, accurately and precisely.

Figure 1 shows the control charts for the methods where the areas of the peaks stay between the established limits every time. The upper and lower limits for the control chart were established at ±2SD (warning limit) and ±3SD (action limit) of this value, taking as SD the value obtained from the variance of the analytical method. To calibrate the UPLC system and monitor its performance, we analyzed an antiTBs solution sample daily as standard. The standards were prepared at 20 µg/mL for INH and PZA, and 25 µg/mL for RFP. The estimated area for these standard concentrations were 638,506 µV·s with an RSD of 2.9%, 703,609 µV·s with an RSD of 4.6%, and 1.053,683 µV·s with an RSD of 6.6% for INH, PZA, and RFP, respectively. The chromatographic conditions (flow-rate, relative mobile-phase composition) and column performance were checked, especially the tailing factor and column efficiency. When it was required, corrective action was taken.

### 2.2. API Stability Studies

A study of the antiTBs’ stability at a temperature of 37 ± 0.1 °C and different pH conditions representing the digestive tract (1.25; 3.0; 6.3; 7.4) was performed in a thermostatic bath (Stuart Scientific SBS30, London, UK). According to pharmacopoeia recommendations, the vehicle used as an artificial gastric juice was prepared with sodium chloride, pepsin, purified water, and hydrochloric acid. The final pH was adjusted with sodium hydroxide to 1.25 or 3.0 [20]. To simulate the intestinal tract, an artificial intestinal juice was prepared with sodium hydroxide, pancreas powder, purified water, and dihydrogen potassium phosphate, finally adjusted to pH 6.3 or 7.4 with sodium hydroxide [20]. The pHs were adjusted using a pH meter at 25 °C. For each pH, the stability was studied for each API individually and a fixed dosage in solution, using the minimum doses currently recommended by WHO: 7, 30, and 10 mg/kg/day for INH, PZA, and RFP, respectively [21,22]. The concentrations of antiTBs tested were calculated considering the normalized gastric volume of 40 mL for an infant with 10 kg body weight [22,23].

At different times, the samples (by triplicate) were diluted with MilliQ water to a concentration within the linear interval of concentrations studied (17.5 µg/mL for INH and PZA, and 25 µg/mL for RFP) and analyzed immediately by UPLC. The evolution of the chromatographic areas was fitted to different kinetic orders to calculate the shelf life at which 5% of the initial dose is degraded (t5%).

### 2.3. RFP:HPBCD Solubility Profiles

An excess of RFP for a 5 mL of volume dose was added (75 mg) to amber glass vials of 8 mL of capacity. Different solutions of HP-β-cyclodextrin (HPBCD) were prepared between 0–0.054 M in a pH 7.4 and 8.0 phosphate buffered [24]. 5 mL of every HPBCD solutions were placed in every vial in duplicate. After this, the vials were sealed and kept for three days in agitation at 375 rpm in an orbital agitator (Heidolph, Schwabach, Germany) at 25 ± 0.1 °C in an oven (J.P. Selecta Medilow, Madrid, Spain).

After three days, the samples were filtered through a syringe filter with a pore size of 0.45 μm (Macherey-Nagel, Dueren, Germany), diluted with MilliQ water, and analyzed by UPLC.

### 2.4. RFP:HPBCD Characterization

Characterization of RFP and HPBCD interaction was made by differential scanning calorimetry (DSC) and infrared spectroscopy (IR) techniques. All solid products were lyophilized from the solution of solubility profile samples.

### 2.5. General Standard Operating Procedure (SOP)

Three different formulations were elaborated with a pediatric dose combination of INH and RFP to cover the continuation phase of TB treatment. The composition of each formulation is shown in Table 1. The minimum dose currently recommended by WHO for INH and RFP [22,23] was used. Doses were calculated for 5 kg of body weight of a child aged three to five months according to Child Growth Standards published by OMS [25]. All the excipients added to these formulations are accepted for pediatrics and have been used below the limits for these populations as EMA recommends. HPBCD proportion (5.6% *p*/*v*, corresponding with a concentration of 0.040 M) used is under the threshold of 200 mg/kg/day given by EMA for children under two years [26,27]. An antioxidant as ascorbic acid was added in different proportions at F2 and F3 in order to improve the RFP stability under presence of INH [28]. The proportions used were the minimum and maximum concentration to be used as an antioxidant [29], and these values are under the recommended dietary allowances for children under six months [30]. At least two batches of each formulation were prepared.

Then, formulations were elaborated according to the following SOP with combined doses of RFP and INH of 10 mg/mL and 7 mg/mL, respectively:The buffer phosphate vehicle at pH 8.0 is prepared [24]. For 1 L: 55 mL solution A (908 mg KH_2_PO_4_ in 100 mL purified water) is added to 945 mL solution B (11.9 g Na_2_HPO_4_·2H_2_O in 100 mL purified water).HPβCD solution is prepared (5.6 % *p*/*v*).RFP dose (1.0 g) is weighted and transferred to a 100 mL glass amber bottle.100 mL of HPβCD solution is transferred to glass amber bottle, a magnetic stirrer is introduced, and the bottle is closed.Preparation is kept in constant shaking during 24 h at 25 °C.Only for F2 and F3, 10 mg, or 100 mg of ascorbic acid, respectively, is weighted and transferred to the 100 mL glass amber bottle. Preparation is kept in constant shaking during 15 min at 25 °C.INH dose (0.7 g) is weighted and transferred to the 100 mL glass amber bottle.Preparation is kept in constant shaking during 30 min at 25 °C.The magnetic stirrer is removed, and the bottle is closed with a dispenser closed.

### 2.6. Formulation Stability Study

Stability for each formulation was checked at 5 °C temperature (Fridge-stove P-selecta Welidow type, Madrid, Spain) following the International Conference Harmonization (ICH) guidelines [31]. Samples (5 mL) from each formulation were taken every 1, 3, 7, and 14 days in duplicate. Every sample was filtered through a syringe filter with a pore size of 0.45 μm (Macherey-Nagel, Dueren, Germany) and diluted with purified water to quantify API content by UPLC.

## 3. Results

### 3.1. API Stability Studies

Table 2 shows the remaining percentage at 0.75 h, the mean gastric residence time for aqueous solutions in infants [32], of every API tested individually or as one combined dose at pH interval between 1.25 and 7.4 and 37 °C. As can be seen, INH and PZA are above 95% of initial dose at all pH conditions at single or combined. RFP maintains its initial percentage above 95% at pH 6.3 and 7.4 for both types of samples, individual or combined. However, with a more acidic pH (1.25 and 3.0), the initial percentage is below 95% in both types of samples.

Depending on the type of samples, the degradation kinetics observed are different, even in the case of different pHs within the same sample type (see Table 3). The best fit to a degradation kinetics order for each API analyzed (individual or in combination) is shown. The time at which 5% (t5%) of the initial drug dose was degraded was calculated through this degradation kinetics. As can be observed, INH and PZA have a t5% above this value at every pH when administered individually or as one combined dose. The individual dose, RFP, remains above 95% of the initial value for more than 1 h. However, in combination at pH 1.25 and 3.0, it is degraded by more than 5% before the stomach empties.

### 3.2. RFP:HPBCD Solubility Profile

As seen in Figure 2, the solubility of the drug increases linearly as a function of the HPBCD concentration, showing a water-soluble complex (A_L_ type). Apparent 1:1 stability constant (kc) of the HPBCD complexes was calculated from the slope of the straight portion of the phase solubility diagram using the equation [33]:k_C_ = slope/S_0_ (1 − slope)(1)
where S_0_ is the intrinsic solubility of the drug.

The complexing efficiency (CE) was calculated from the slope of the phase-solubility diagram using the equation [33]:CE = slope/(1 − slope)(2)

The slope of the straight line, calculated for 1:1 RFP:HPBCD complexes in the 0–0.017 M RFP concentration range vs in 0–0.054 M HPBCD concentration range, has a value less than unity for every pH studied. The intercept (or intrinsic solubility value) is equal to 3.14 mM and 3.71 mM for pH 7.4 and 8.0, respectively. kc and CE values are: for pH 7.4, 83 M^−1^ and 0.261, respectively, and for pH 8.0, 94 M^−1^ and 0.351, respectively. Therefore, the complexation has a kc and CE values higher at a more alkaline pH.

### 3.3. RFP:HPBCD Characterization

The formation of RFP:HPBCD was confirmed by DSC and IR techniques. As Figure 3a shows, DSC thermogram of RFP exhibited one peak at 200 °C, which corresponds with RFP melting point. HPBCD present characteristic DSC thermogram described in the literature [14]. For RFP:HPBCD thermogram the characteristic RFP peak at the same temperature but with less intensity, suggesting RFP molecule might be enclosed in the HPBCD inner cavity. Similarly, as shown in Figure 3b, the RFP:HPBCD IR spectrum was similar to HPBCD, indicating that there is molecular interaction between RFP and HPBCD.

### 3.4. Formulation Stability Studies

Due to the results obtained in the previous section, the selected vehicle to prepare the formulations was phosphate buffer at pH 8.0.

F1, F2, and F3 initial doses were: 10.5 ± 0.4, 10.3 ± 0.4, and 9.03 ± 0.2 mg/mL and 6.5 ± 0.1, 7.3 ± 0.0, and 7.0 ± 0.3 mg/mL of RFP, and INH, respectively (see Table 4). Then, the target doses were achieved when the preparation of formulations were finished. After three days in storage at 5 °C, the remaining percentage was above 95% for INH at F1 and F3. At the same storage time, RFP percentage was above 90% at F3. The remaining content and pH of every formulation was progressively decreased until 15 days of storage. pH decreases, Table 4, from the initial value of 8.0 to values of 7.5 ± 0.06, 7.1 ± 0.05 and 7.2 ± 0.01 for F1, F2, and F3, respectively, with no statistically significant differences (*p* > 0.05) between days of storage for any formulation studied.

## 4. Discussion

If an API has a t5% below 0.75 h, the mean gastric residence time for aqueous solutions in paediatrics means that 5% of the initial dose is degraded before it can be absorbed in the intestine. The only API that shows t5% below 0.75 h at pH 1.25 and 3.0 in combination is RFP. The kinetic constants obtained show that degradation dependence with time and initial concentration can be not the same along the pH interval studied, or if the samples are individual or combined. Our results confirm studies carried out on adults. RFP at paediatric doses is the most unstable antiTB during its residence in the stomach before passing to the duodenum for absorption. This instability increases when administered as a combined dose, probably due to the interaction with INH when administered in combination below pH 2, among other reasons [34,35].

RFP forms inclusion complexes with HPBCD of different solubilities in phosphate buffer at pH 7.4 and 8.0, increasing its solubility 5.3 times at 25 °C. At pH 8.0, CE is 0.351, a value which is greater than for pH 7.4. This means that assuming 1:1 RFP:HPBCD complex formation, that on an average only about one out of every three HPBCD molecules in solution (unlike four for pH 7.4) are forming a water-soluble complex with the poorly soluble RFP. These results are similar to the obtained by Tewes et al. for lung nebulization, the use of HPBCD allows higher RFP dosing into the lung by increasing the RFP apparent solubility [13].

The introduction of HPBD in the preparation of a formulation of RFP in combination with INH makes it possible to prepare a liquid oral solution to be administered in paediatrics. The initial pH of the formulations must be 8.0 to obtain a better HPBCD:RFP complex and be able to solubilize the paediatric dose of RFP pH of the formulations decreases after the incorporation of all components of the formulation to values close to 7.4, which is the pH of maximum stability of the in vitro gastrointestinal conditions previously studied. The remaining doses are maintained above 90% of the initial dose for both APIs when ascorbic acid is used as an antioxidant in a proportion of 0.1% w/v during three days of storage at 5 °C of storage. Our results are in accordance with the results obtained for ascorbic acid by Rajaram et al. [28] and Aliabadi et al. [36]. The first studied its effect in the stability and pharmacokinetics of RFP in the presence of INH in the stomach acid environment [28]. They demonstrate that this co-administration with these antiTBs in combination protect RFP from degradation but not in an acid environment. Aliabadi et al. incorporated ascorbic acid at 0.1% w/v for the preparation of a mercaptopurine oral suspension, and increased the suspensions’ shelf life at room temperature [36].

## 5. Conclusions

AntiTBs for first-line treatment in paediatrics can be administered orally as fixed combined doses (INH and RFP) formulated in liquid forms. The combined doses as liquid oral formulation must be formulated with an alkaline buffer to increase the stability of RFP. This would improve the absorption in the duodenum and its higher bioavailability.

The RFP:HPBCD complex formation at pH 8.0 and 25 °C offers better stability and complexing efficiency than at pH 7.4, improving RFP solubility and enabling the preparation of a liquid solution in which the required initial doses of RFP and INH are dissolved.

The incorporation of ascorbic acid as antioxidant at 0.1% w/v at F3, improves the stability of INH and RFP during the formulation storage at 5 °C for at least three days.

Future assays are required and they should be carried out in different ways. It is necessary to test the RFP:HPBCD solubility profile in presence of INH and ascorbic acid and characterize the inclusion complex formed, as well as how to prolong F3’s shelf life for at least 14 days in order to use it in the continuation phase of TB treatment. Finally, it would be necessary to study the incorporation of PZA to the combination in solution of INH and RFP in order to use in the intensive phase of TB treatment and to improve the stability of the APIs in the formulation at different temperatures as the International Conference of Harmonization indicates.

## Figures and Tables

**Figure 1 pharmaceutics-12-00195-f001:**
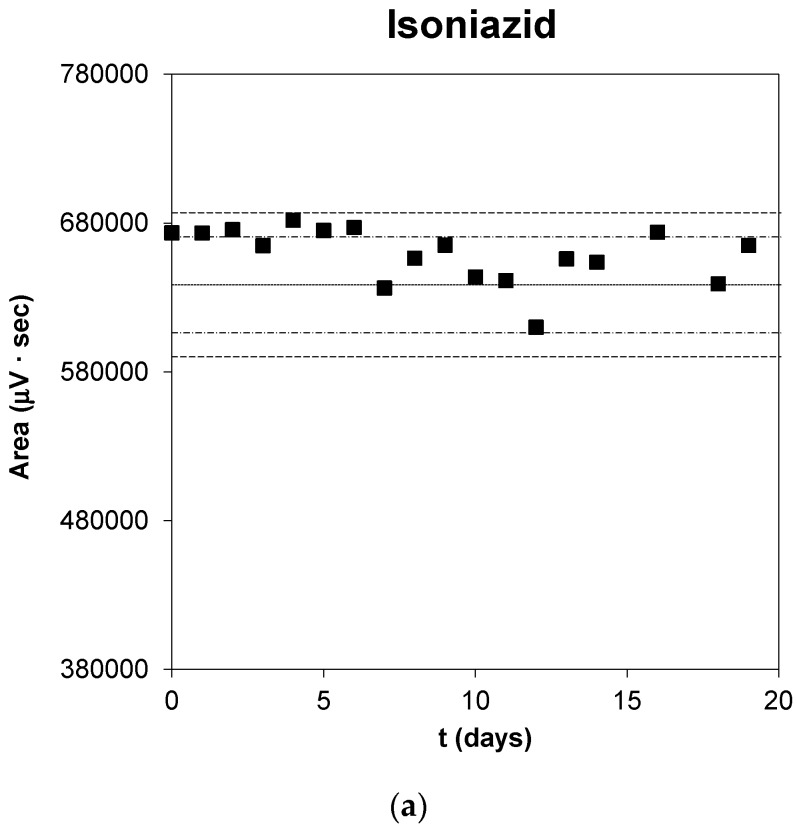
Control chart of the UPLC method for each antiTB. The estimated areas, warning (±2SD), and action limit (±3SD) are marked with a dashed, dashed, and dotted, and more dashed line respectively.

**Figure 2 pharmaceutics-12-00195-f002:**
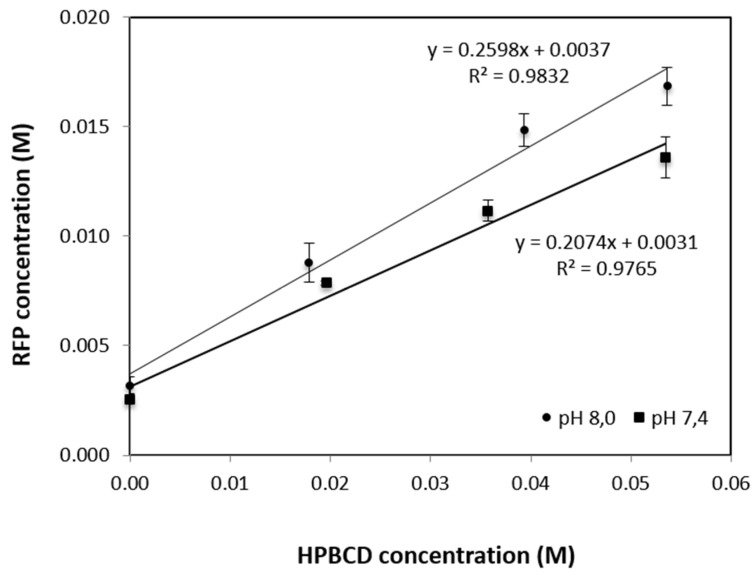
RFP:HPBCD solubility profile at pH 7.4 and 8.0 at 25 °C.

**Figure 3 pharmaceutics-12-00195-f003:**
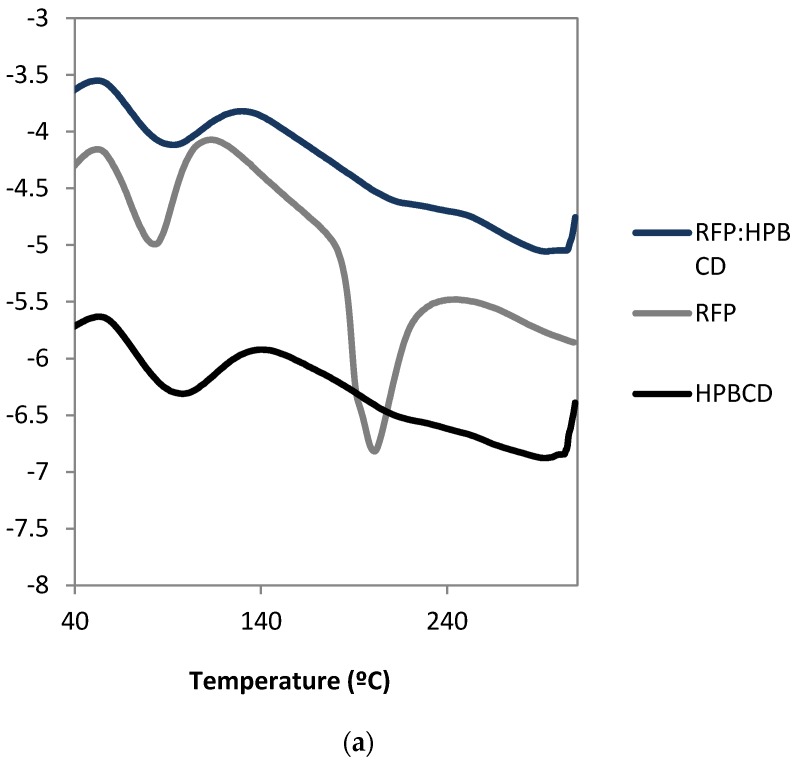
Differential scanning calorimetry (**a**) and infrared spectra results (**b**) of lyophilized solution samples from solubility profile assay.

**Table 1 pharmaceutics-12-00195-t001:** Composition of every formulation studied. INH = Isoniazid; RFP = Rifampicin; HPBCD = hydroxypropyl-β-cyclodextrin.

Products	F1	F2	F3
INH (g)	0.7	0.7	0.7
RFP (g)	1.0	1.0	1.0
HPBCD (% *w*/*v*)	5.6	5.6	5.6
Ascorbic acid (% *w*/*v*)	-	0.01	0.1
Phosphate buffer pH 8.0	100 mL	100 mL	100 mL

**Table 2 pharmaceutics-12-00195-t002:** Percentage of INH, PZA and RFP remaining (R%) after 0.75 h storage at different pH and 37 °C. SD = Standard Deviation; INH = Isoniazid; PZA = Pyrazinamide; RFP = Rifampicin.

**INH**	**INDIVIDUAL**		**COMBINED**	
**pH**	**R (%)**	**SD**	**R (%)**	**SD**
1.25	100.1	2.84	97.2	1.08
3.0	95.7	1.10	99.8	0.00
6.3	98.3	5.25	103.2	5.22
7.4	104.4	4.00	99.8	2.0
**PZA**	**INDIVIDUAL**		**COMBINED**	
**pH**	**R (%)**	**SD**	**R (%)**	**SD**
1.25	112.6	3.03	99.8	0.20
3.0	99.2	1.76	99.8	0.00
6.3	95.0	2.14	96.2	1.99
7.4	97.4	1.20	99.7	4.10
**RFP**	**INDIVIDUAL**		**COMBINED**	
**pH**	**R (%)**	**SD**	**R (%)**	**SD**
1.25	94.7	0.18	89.4	2.28
3.0	89.0	0.82	88.7	3.90
6.3	96.6	6.25	95.05	12.8
7.4	103.4	1.40	100.8	3.50

**Table 3 pharmaceutics-12-00195-t003:** Kinetic orders to calculate the shelf life at which 5% of the initial dose is degraded (t5%). r: correlation coefficient. INH = Isoniazid; PZA = Pyrazinamide; RFP = Rifampicin.

**INH**	**INDIVIDUAL**			**COMBINED**		
**pH**	**Order**	**t5% (h)**	**r**	**Order**	**t5% (h)**	**r**
1.25	2	6.6	0.988	0	7.6	0.990
3.0	1	6.2	0.900	2	6.6	0.691
6.3	2	3.9	0.913	2	11.2	0.922
7.4	0	18.0	0.995	2	9.4	0.97
**PZA**	**INDIVIDUAL**			**COMBINED**		
**pH**	**Order**	**t5% (h)**	**r**	**Order**	**t5% (h)**	**r**
1.25	2	96.1	0.970	0	44.6	0.922
3.0	2	8.2	0.880	-	> 24	-
6.3	2	173.9	0.941	-	> 24	-
7.4	2	78.7	0.843	1	27.8	0.999
**RFP**	**INDIVIDUAL**			**COMBINED**		
**pH**	**Order**	**t5% (h)**	**r**	**Order**	**t5% (h)**	**R**
1.25	1	1.2	0.995	1	0.5	0.975
3.0	1	2.1	0.991	1	0.7	0.994
6.3	1	2.5	0.952	2	1.8	0.940
7.4	1	7.8	0.978	1	5.8	0.995

**Table 4 pharmaceutics-12-00195-t004:** Initial doses, pH, and percentages of INH and RFP remaining (R%) for 14 days storage of F1, F2, and F3 at pH 8.0 and 5 °C. SD = Standard Deviation; INH = Isoniazid; RFP = Rifampicin.

**INH**			**t (days)**							
	**Initial Dose (mg/mL)**	**SD**	1		3		7		14	
			R (%)	SD	R (%)	SD	R (%)	SD	R (%)	SD
F1	6.5	0.1	100	0.00	96.6	2.37	92.6	6.27	92.3	10.9
F2	7.3	0.0	100	0.00	86.4	1.00	83.7	3.29	76.8	1.25
F3	7.0	0.3	100	0.00	95.0	0.03	89.1	0.68	83.4	0.26
**RFP**	**Initial Dose (mg/mL)**	**SD**	**t (days)**							
			1		3		7		14	
			R (%)	SD	R (%)	SD	R (%)	SD	R (%)	SD
F1	10.5	0.4	100	0.00	75.5	1.87	68.3	3.92	45.3	4.76
F2	10.3	0.4	100	0.00	88.1	0.21	61.0	3.80	41.3	0.82
F3	9.03	0.2	100	0.00	91.6	7.86	64.2	4.92	48.0	2.56
**pH**			**t (days)**							
			1		3		7		14	
			pH	SD	pH	SD	pH	SD	pH	SD
F1			7.47	0.03	7.45	0.07	7.53	0.01	7.47	0.06
F2			7.35	0.01	7.53	0.01	7.25	0.04	7.11	0.05
F3			7.24	0.00	7.28	0.00	7.20	0.01	7.19	0.01

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
