# Peer review of "Stability Study of Isoniazid and Rifampicin Oral Solutions Using Hydroxypropyl-Β-Cyclodextrin to Treat Tuberculosis in Paediatrics"

_pharmaceutics, 2020, doi:10.3390/pharmaceutics12020195_

Round 1

Reviewer 1 Report

This study developed a combined dose liquid formulation of antiTB for
oral use in paediatrics with good stability and solubility. The work is
of good practical significance. But careful revision is needed as suggested as follows.
1. In Table 1, only INH and RFP were listed, However, in the title of Table 4, “Initial doses, pH and percentages of INH, PZA and RFP remaining (R%) during 14 days storage”. Was PZA included in the formulation?
2. As stated at Section 2.5, stability was tested for 14 days at 5 ºC. However, the title of Table 4, "Initial doses, pH and percentages of INH, PZA and RFP remaining (R%) during 14 days storage of F1, F2 and F3 at pH 8.0 and 25 ºC. " Which temperature was tested, 5 ºC or 25 ºC? Was the testing time duration of only 14 days long enough for predicting shelf life? In addition, the R% of RFP fell to below 50% by 14 days. The stabiliby seemed poor. Please explain.
3. The formation of inclusion of RFP should be characterized with more details.
4. The Conclusion part should be more concise.

Author Response

Manuscript ID: pharmaceutics-716111

Thank you very much for your email with the comments from the reviewers. We appreciate their thoughtful and useful comments. We apologize that the initial version of the manuscript was not sufficiently elaborated. The whole manuscript has been carefully revised taking into account the particular comments of the reviewers. All their concerns have been carefully addressed. Changes in the manuscript are highlighted in red. Please find below a point-by-point answer to the reviewers´ comments. In addition, the English and the edition have been revised throughout the manuscript.

Reviewer: 1

This study developed a combined dose liquid formulation of antiTB for oral use in paediatrics with good stability and solubility. The work is of good practical significance. But careful revision is needed as suggested as follows.
1. In Table 1, only INH and RFP were listed, However, in the title of Table 4, “Initial doses, pH and percentages of INH, PZA and RFP remaining (R%) during 14 days storage”. Was PZA included in the formulation?

ANSWER: No, PZA was not included in the formulation. There was an error in the title of table 4 which has been changed. In order to follow the same criteria with all tables, tables 1 and 4 have been modified in the corrected text.

  1. As stated at Section 2.5, stability was tested for 14 days at 5 ºC. However, the title of Table 4, "Initial doses, pH and percentages of INH, PZA and RFP remaining (R%) during 14 days storage of F1, F2 and F3 at pH 8.0 and 25 ºC. " Which temperature was tested, 5 ºC or 25 ºC? Was the testing time duration of only 14 days long enough for predicting shelf life? In addition, the R% of RFP fell to below 50% by 14 days. The stabiliby seemed poor. Please explain.

ANSWER: As section 2.5 shows, formulation stability study was tested at 5ºC. There was a mistake in the title of table 4. It has been now corrected. Testing time duration was 14 days because the percentage remaining for both APIs were below 90%. This study adeals with the initial approximation to a liquid stable formulation combining INH and RFP through the addition to the formulation of hydroxypropyl-β-cyclodextrin and ascorbic acid. These excipients enabled us to solubilise the paediatric doses at one day following elaboration and store them during 3 days at 5 ºC above 90% for F3 formulation.

  1. The formation of inclusion of RFP should be characterized with more details.

ANSWER: Yes, we agree with the reviewer. As pointed out in the conclusion section, future assays should be carried out in order to characterize the inclusion complex formed with more detail in presence of INH and ascorbic acid. Our research group is presently testing the inclusion complex at different pHs by Differential Scanning Calorimetry (DSC), Infrared spectroscopy (IR) and nuclear magnetic resonance (NMR), and these results will be included in another manuscript. We have included a new section “RFP:HPBCD characterization” in the material and methods and results sections respectively, and figure 3 with the aim of confirming by DSC and IR the formation of the inclusion complex.

  1. The Conclusion part should be more concise.  

ANSWER: The conclusions have been changed in order to be more concise.

Yours sincerely,

Ana Santoveña

Reviewer 2 Report

The manuscript by Santovena-estevez et al describes a study on the stability of ora solutions of Isoniazid and rifampicin in simulated gastric conditions when complexed with cyclodextrin. The study is reasonably well described and with a few minor changes could be suitable for publication.

1) the analysis of previous literature regarding the study of complexes with cyclodextrin should be expanded, to better present what is known regarding the stability and complexation kinetics.

2)while there is a reasonable analysis of complexation, it is not adequately described the kinetics of desorption from the cyclodextrin chelator. Even though a complete study in this sense is probably beyond the scope of ths work, the authors should at least discuss potential incunveniences due to this complexation.

3) it is not completely clear to me the strange kinetics behavior of PZA that shows alternating kinetics constants. The authors should better discuss this behavior or improve the quaity of their data (note that the two t5% at pH 3.0 and 7.4 hav also the poorest r. Further replicates could allow improving these data).

Author Response

Manuscript ID: pharmaceutics-716111

Thank you very much for your email with the comments from the reviewers. We appreciate their thoughtful and useful comments. We apologize that the initial version of the manuscript was not sufficiently elaborated. The whole manuscript has been carefully revised taking into account the particular comments of the reviewers. All their concerns have been carefully addressed. Changes in the manuscript are highlighted in red. Please find below a point-by-point answer to the reviewers´ comments. In addition, the English and the edition have been revised throughout the manuscript.

Reviewer 2

The manuscript by Santovena-estevez et al describes a study on the stability of ora solutions of Isoniazid and rifampicin in simulated gastric conditions when complexed with cyclodextrin. The study is reasonably well described and with a few minor changes could be suitable for publication.

1) the analysis of previous literature regarding the study of complexes with cyclodextrin should be expanded, to better present what is known regarding the stability and complexation kinetics.

ANSWER: We agree with the reviewer. An analysis of previous literature regarding the study of complexes with hydroxypropyl-β-cyclodextrin has been now included in the introduction section.

2) while there is a reasonable analysis of complexation, it is not adequately described the kinetics of desorption from the cyclodextrin chelator. Even though a complete study in this sense is probably beyond the scope of ths work, the authors should at least discuss potential incunveniences due to this complexation.

ANSWER: Yes, the kinetics of desorption are being checked by different analytical techniques such as Differential Scanning Calorimetry, Infrared spectroscopy, etc, and these results will be included in another manuscript in which a more detailed study is carried out on the molecular association of RFP with HPBCD in presence of other APIs (INH and PZA) and excipients (ascorbic acid, etc).  

3) it is not completely clear to me the strange kinetics behavior of PZA that shows alternating kinetics constants. The authors should better discuss this behavior or improve the quaity of their data (note that the two t5% at pH 3.0 and 7.4 hav also the poorest r. Further replicates could allow improving these data).

ANSWER: We agree with the reviewer. The kinetic behaviour of PZA showed different kinetic orders. These results are the average of two different stability studies taking samples in each one by triplicate. The best fit to different kinetic orders was selected for every pH. The kinetic behaviour showed in table 3 is the best result obtained although the kinetics constants were different. It must be into account that pH of the medium changes the ionic status of the API and the degradation dependence with time and initial concentration can be not the same along the pH interval studied. This aspect is now indicated in the corrected text.

Yours sincerely,

Ana Santoveña

Reviewer 3 Report

The submitted work presents results of a stability study of a solution of isoniazide and rifampicin for oral administration in the presence or absence of ascorbic acid, added as stabilizer and cyclodextrin, added as solubility enhancer for rifampicin. The developed solution formulation is proposed potentially for administration to children suffering from tuberculosis. A third drug, Pyrazinamide is also included in the stability studies but not in the finally tested formulations.

The work is clearly presented but cannot be considered completed yet. Stability studies a) including all three drugs b) isoniazide and PZA and c) rifampicin and PZA are needed to address thoroughly the question of stability since all the above three above regiments appear in the medical literature.  Besides, the above added data will answer the question which of the above drug combinations can be applied in the same solution formulation for therapy.

Finally, the authors should seek help with the language presentation and I hope my comments below will help a little with this issue.

Specific comments

Line 21 – Replace analyze with examine

L. 23-26 – The sentence ‘The active pharmaceutical ingredients stability under in vitro paediatric digestive pH conditions have been studied when administered as individual or fixed combined paediatric dosages to determine the pH of maximum stability’ is not unerstood. Unfortunately this sentence contains the objective of the work, and hence it must be split into two or three smaller, so as to convey the aim of the work clearly.

L. 31. – ‘Three days storage …’ What is the intended use of the formulation? Such a short shelf life excludes commercialization.

L. 52. - Name the excipients added in the tablets that are not suitable for children.

L. 81. – ‘… a solubility diagram was used …’. Such diagram is not found in the submission.

L. 89. – Materials and methods. Give full names and suppliers for the drugs, type and supplier of cyclodextrin in this paragraph.

L. 121. Suppliers must be reported in the Materials and methods.

L. 164. What is the amount of HPCD in the final formulation per 100 ml?

L. 253, Add ‘in’ before accordance.

Author Response

Manuscript ID: pharmaceutics-716111

Thank you very much for your email with the comments from the reviewers. We appreciate their thoughtful and useful comments. We apologize that the initial version of the manuscript was not sufficiently elaborated. The whole manuscript has been carefully revised taking into account the particular comments of the reviewers. All their concerns have been carefully addressed. Changes in the manuscript are highlighted in red. Please find below a point-by-point answer to the reviewers´ comments. In addition, the English and the edition have been revised throughout the manuscript.

Reviewer 3

The submitted work presents results of a stability study of a solution of isoniazide and rifampicin for oral administration in the presence or absence of ascorbic acid, added as stabilizer and cyclodextrin, added as solubility enhancer for rifampicin. The developed solution formulation is proposed potentially for administration to children suffering from tuberculosis. A third drug, Pyrazinamide is also included in the stability studies but not in the finally tested formulations.

The work is clearly presented but cannot be considered completed yet. Stability studies a) including all three drugs b) isoniazide and PZA and c) rifampicin and PZA are needed to address thoroughly the question of stability since all the above three above regiments appear in the medical literature.  Besides, the above added data will answer the question which of the above drug combinations can be applied in the same solution formulation for therapy.

Finally, the authors should seek help with the language presentation and I hope my comments below will help a little with this issue.

Specific comments

Line 21 – Replace analyze with examine

ANSWER: It has been replaced “analyses” with “examines”.

  1. 23-26 – The sentence ‘The active pharmaceutical ingredients stability under in vitro paediatric digestive pH conditions have been studied when administered as individual or fixed combined paediatric dosages to determine the pH of maximum stability’ is not unerstood. Unfortunately this sentence contains the objective of the work, and hence it must be split into two or three smaller, so as to convey the aim of the work clearly.

ANSWER: We agree with the reviewer. This sentence has been now splitted to better understood.

  1. 31. – ‘Three days storage …’ What is the intended use of the formulation? Such a short shelf life excludes commercialization.

ANSWER: Yes, this is a short shelf life. Our research group are studying now the inclusion complex formed with more detail in presence of INH and ascorbic acid in order to prolong the stability of the formulation. This is study is an initial approximation to the final goal at least about 15 days.

  1. 52. - Name the excipients added in the tablets that are not suitable for children.

ANSWER: The excipients not suitable for children are now included in the corrected text.

  1. 81. – ‘… a solubility diagram was used …’. Such diagram is not found in the submission.

ANSWER: Yes, the solubility diagram is showed in figure 2.

  1. 89. – Materials and methods. Give full names and suppliers for the drugs, type and supplier of cyclodextrin in this paragraph.

ANSWER: The full names and suppliers for the drugs, and the type and supplier of cyclodextrin is now included in the corrected text.

  1. 121. Suppliers must be reported in the Materials and methods.

ANSWER: We agree with the reviewer. The suppliers are now included at the beginning of Material and Methods section.

  1. 164. What is the amount of HPCD in the final formulation per 100 ml?

ANSWER: It is previously prepared a solution of 5.6 % w/v. Then there is 5.6 g per 100 ml.

  1. 253, Add ‘in’ before accordance.

ANSWER: Thank you, “in” has been now added in the corrected text.

Yours sincerely,

Ana Santoveña

Round 2

Reviewer 1 Report

Most questions have been answered. However, I am still concerned with the stability. As stated by the author, "Testing time duration was 14 days because the percentage remaining for both APIs were below 90%. " Would such stability be adequate for practical application?

Author Response

Manuscript ID: pharmaceutics-716111

Thank you for your email with the comments of the reviewers. Please find below point-by-point answers to the reviewers´ comments. All their concerns have been carefully addressed. Changes in the manuscript are highlighted in red. Please find below a point-by-point answer to the reviewers´ comments. In addition, the English and the edition have been revised again throughout the manuscript.

Reviewer: 1

Most questions have been answered. However, I am still concerned with the stability. As stated by the author, "Testing time duration was 14 days because the percentage remaining for both APIs were below 90%. " Would such stability be adequate for practical application?

ANSWER: No, a percentage that remains below 90% of any API indicates degradation. But up to 3 days the percentage of both APIs is above 90% at F3. With the aim of prolonging the shelf life of the formulation for at least 14 days we are now studying the evolution of the complexation process during storage.  The main achievement of this study is to solubilize and maintain the stability of the pediatric doses of both APIs for at least 3 days. This allows us to make extemporaneous preparations in the last instance if it is not possible to prolong the life of F3 for at least 14 days. This has now been included in the conclusions section of the corrected text.

Yours sincerely,

Ana Santoveña

Reviewer 3 Report

Authors corrected the points that were raised in the specific comments but did not respond to the point of why stability of formulations containing all three drugs are not presented. In particular they should provide adequate explanations of why they think that isoniazide/PZA, rifampicin/PZA and the triple combination of isoniazide/rifampicn/PZA do not merit investigation and why they focused only on the rifampicin/isoniazide? 

Author Response

Manuscript ID: pharmaceutics-716111

Thank you for your email with the comments of the reviewers. Please find below point-by-point answers to the reviewers´ comments. All their concerns have been carefully addressed. Changes in the manuscript are highlighted in red. Please find below a point-by-point answer to the reviewers´ comments. In addition, the English and the edition have been revised again throughout the manuscript.

Reviewer 3

Authors corrected the points that were raised in the specific comments but did not respond to the point of why stability of formulations containing all three drugs are not presented. In particular they should provide adequate explanations of why they think that isoniazide/PZA, rifampicin/PZA and the triple combination of isoniazide/rifampicn/PZA do not merit investigation and why they focused only on the rifampicin/isoniazide? 

ANSWER: Our final objective is to prepare a triple combination INH/PZA/RFP used in the intensive phase of TB treatment. We have started studying the simplest combination used in the continuation phase of the TB treatment and this is INH/RFP. Combinations of INH/PZA or PZA/RFP are not needed in the different phases of the TB treatment. This has now been clarified in the conclusions section of the corrected text.   

Yours sincerely,

Ana Santoveña